# A Multi–Membrane System to Study the Effects of Physical and Metabolic Interactions in Microbial Co-Cultures and Consortia

**Jennifer R. Oosthuizen** [1], **Rene K. Naidoo-Blassoples** [1], **Debra Rossouw** [1], **Robert Pott** [2] and **Florian F. Bauer** [1,*]

[1] Department of Agrisciences, South African Grape and Wine Research Institute, Stellenbosch University, Stellenbosch 7600, South Africa; jeno@sun.ac.za (J.R.O.); rknaidoo@sun.ac.za (R.K.N.-B.); debra@sun.ac.za (D.R.)
[2] Department of Process Engineering, Stellenbosch University, Stellenbosch 7600, South Africa; Rpott@sun.ac.za
[*] Correspondence: fb2@sun.ac.za

**Abstract:** Continuous cell-to-cell contact between different species is a general feature of all natural environments. However, almost all research is conducted on single-species cultures, reflecting a biotechnological bias and problems associated with the complexities of reproducibly growing and controlling multispecies systems. Consequently, biotic stress due to the presence of other species remains poorly understood. In this context, understanding the effects of physical contact between species when compared to metabolic contact alone is one of the first steps to unravelling the mechanisms that underpin microbial ecological interactions. The current technologies to study the effects of cell-to-cell contact present disadvantages, such as the inefficient or discontinuous exchange of metabolites when preventing contact between species. This paper presents and characterizes a novel bioreactor system that uses ceramic membranes to create a "multi-membrane" compartmentalized system whereby two or more species can be co-cultured without the mixing of the species, while ensuring the efficient sharing of all of the media components. The system operates continuously, thereby avoiding the discontinuities that characterize other systems, which either have to use hourly backwashes to clean their membranes, or have to change the direction of the flow between compartments. This study evaluates the movement of metabolites across the membrane in co-cultures of yeast, microalgae and bacterial species, and monitors the movement of the metabolites produced during co-culturing. These results show that the multi-membrane system proposed in this study represents an effective system for studying the effects of cell-to-cell contact in microbial consortia. The system can also be adapted for various biotechnological purposes, such as the production of metabolites when more than one species is required for such a process.

**Keywords:** membrane; co-culture; cell contact; yeast; algae; cell-contact

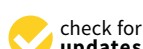

## 1. Introduction

Microbial species do not exist in isolation. The existing research, however, largely focuses on the characterisation of species in isolation, and most biotechnological applications also focus on single-species cultures. Microbial monocultures can be beneficial due to their predictability and ease of control, but they have significant limitations which include sensitivity to environmental perturbation and contamination, as well as limited metabolic capabilities [1,2]. For many biotechnological applications, multispecies systems—either natural or synthetic consortia—would have advantages, including broader metabolic capabilities, robustness to environmental perturbations, resistance to invasive species [3,4], and the ability to withstand periods of stress such as nutrient limitation [5]. Multispecies systems also open a suite of substrate-to-product routes which are not currently available in sequential fermentation systems. However, such multispecies systems are difficult to control and remain largely unpredictable, limiting their applicability.

For this reason, investigations focusing on a better understanding of the ecological dynamics and the interactions between organisms within such systems have become a central

focus of biotechnological and ecological research. The complexity of such interactions, however, presents a tremendous challenge and requires the development of novel tools. In this context, compartmentalised systems, in which different species are kept separate physically, but which allow the movement of metabolites, can be valuable tools in the achievement of better control of multispecies cultures, and in the study of the effects of cell-to-cell contact [6–8]. Understanding the effects of physical cell-to-cell contact on microbial interactions is essential in the characterisation of microbial communities and their functioning. This is commonly achieved by comparing mixed co-cultures to co-cultures which are not in direct cell contact, while still sharing a common culture medium. Although co-cultures have been shown to provide improved functional and metabolic capabilities in comparison to monocultures, the exact mechanism of the improvement is not always clear. Comparing the effects of co-culturing with and without cell contact between microbes can assist us in understanding whether the interaction is mediated through physical contact or metabolic exchange. It has been demonstrated that physical contact between *Saccharomyces cerevisiae* and *Hanseniaspora uvarum* affected the viability of *H. uvarum*, and influenced the metabolic behaviour of the strain [9]. This study was conducted in a double-compartment bioreactor, wherein the species were separated throughout the co-culture; this showed how the use of a mixed starter culture affected non-Saccharomyces yeasts during wine fermentation. An improved understanding of the effects of physical contact between microbial species will ensure their effective exploitation in both industry and research [6,10].

The existing membrane technologies rely primarily on transwell systems, or on systems where the culture is passed through a membrane between two vessels in an alternating manner. Transwell systems rely on the passive diffusion of metabolites between two areas with a membrane partition placed between them. Transwell systems are commonly utilized in research because they are very easy to use, and small volumes make them ideal for large-scale screenings. Such a system was used to study the effects of co-culturing bacterial species on phenotypic heterogeneities [11]. One significant limitation in this experimental set up is that the vertical nature of the passive diffusion in such plates prevents the collection of optical density data. A horizonal membrane placed between two small chambers was an attempted solution for this, but it once again relied only on the passive diffusion of the culture medium through the membrane layer [12]. Such systems are designed for small volumes of culture, usually no more than 5 mL per vessel. These small volumes are not ideal when running analyses that require large sample volumes, and increasing the volume may lead to a decrease in the efficiency of the diffusion of the metabolites in the system. A system used for many co-culture studies today is one whereby pressure on alternating vessel headspaces forces mixing through a hollow fiber membrane module placed between them [13]. This system utilises larger reaction volumes; however, the alternating nature of the system may introduce a bias in the data due to a lack of the continuous flow of the metabolites. The use of 'conditioned' mediums in which a chosen microbial species has been grown and then removed is another method of analysing the effects of cell-to-cell contact in microbial co-cultures [14]. Once the cells of the first species have been removed, the media contains only the cell secretome (which can affect the growth and behaviour of the second species). This method, however, does not account for the possibility of gas exchange (such as the way in which the $CO_2$ production by one species may affect the co-culture), and is limited to unidirectional interactions that do not represent the dynamic nature of co-culturing.

The reliance of the existing technologies on the passive diffusion of molecules or alternating mixing strategies is not efficient, and may not replicate a real-world environment where bi-directional mixing would continuously take place. These methods may also introduce biases in the study of cell-to-cell interactions. Here, we present a custom-designed multi-membrane bioreactor that allows for the co-culturing of two or more microbial species without any cell-to-cell contact. The system addresses most of the drawbacks of the previously described systems, as it allows the continuous and efficient exchange of metabolites and proteins without the disruption of flow, and by not relying on passive

diffusion alone. The system also can be scaled to various sizes, and can in principle be extended to include more than two independent bioreactors. The data show that this system allows for the rapid exchange of metabolites and proteins between two or more bioreactors, ensuring shared medium composition between the reactors while completely separating the microorganisms within these reactors. We validate the system under different conditions and with various microbial species combinations, including co-cultures combining species from vastly different evolutionary origins. The use of systems such as those described here can improve our understanding of the effects of physical cell contact between microbial species, and can allow for their efficient exploitation in both research and industry.

## 2. Materials and Methods

### 2.1. Multi-Membrane Setup

The system is based on parallel pumping circuits for each independently cultured species. The exchange of metabolites and proteins between the culture vessels occurs within stainless steel housings which contain ceramic membranes that provide a large and pressurized interface.

For this system, the membranes were procured from Memcon (Pty) Ltd., Florida, South Africa. These ceramic membranes are cylindrical in shape, can have various pore sizes, and allow for the through-flow of media to prevent cellular build up; pressure forces the media without cells through the membrane and out of the side housing port seen in Figure 1. Liquid culture is circulated through this system using Tygon® Norprene® tubing (Saint-Gobain, Paris, France) with an internal diameter of 8 mm, providing a flow rate of 4 mL/s, using two peristaltic pumps (YZ1515X, Runze Fluid Control Equipment, Nanjing, China). Semi-closed needle valves obtained from RS Components Ltd. (Corby, Northants, UK) create pressure and force the media through the ceramic membrane. Any equivalent equipment could be used for a similar setup of the multi-membrane system. A diagrammatical representation of the new system is shown in Figure 1.

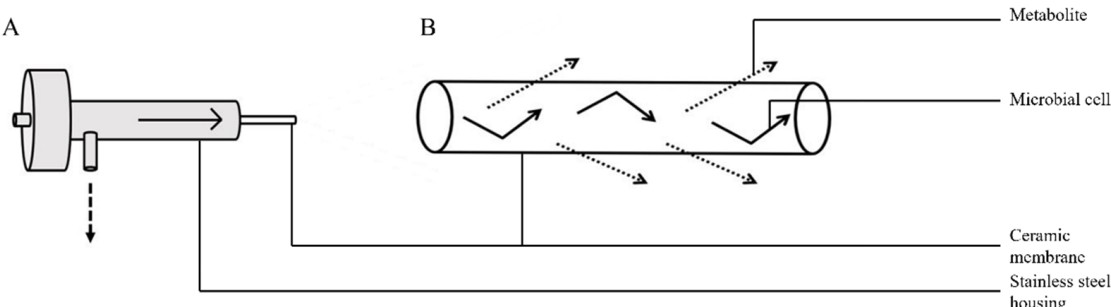

**Figure 1.** Schematic of the stainless-steel housing (**A**) containing a ceramic membrane (**B**). In image B, the solid arrows represent cells too large to pass through the pores in the membrane, and the dotted arrows indicate metabolites small enough to pass through the pores in the membrane. The placement of the membrane housing can be seen within the larger context of the system in Figures 2 and 3.

### 2.2. Membrane Permeability Testing

A variety of metabolites were monitored in the multi-membrane system in order to ensure that the effective mixing of the co-culture medium took place. The system was assembled and initially run with distilled water in one vessel and a mixture of metabolites of known concentration in distilled water in the other (Table 1). These test media were used only to determine the movement of specific substances through the membrane. Amino acids were added to the medium at the specified concentrations for synthetic grape must media [15]. Three representative amino acids at varying initial concentrations in the medium were chosen as a representative dataset. The complete dataset of amino acids tested for can be found in Appendix A. Both ceramic membranes were of size 0.1 μm, and the fully assembled system was autoclaved before use. The system was run at 50 rpm

for eight hours, and 20 mL samples from both vessels were taken at various timepoints during the run. These runs were performed in triplicate, alternating the distilled water in either vessel.

**Table 1.** Metabolites used to test the permeability of the ceramic membrane.

| Metabolite | Concentration (g/L) |
|---|---|
| Glucose | 50 |
| Fructose | 50 |
| Acetic acid | 8 |
| Glycerol | 20 |
| Ethanol | 10 |
| Total protein (BSA and lactalbumin) | 20 |
| Amino acids: | |
| Proline | 0.468 |
| Lysine | 0.013 |
| Alanine | 0.111 |

### 2.3. Nutrient Analysis

The metabolite concentrations were analysed by the centrifugation of 2 mL samples at $10,000 \times g$ for 5 min. The supernatant was removed and stored at $-20\,^\circ$C until the assay was performed according to the protocols provided. Glucose and fructose were measured using a D-Fructose/D-Glucose Assay Kit from Megazyme (Bray, Ireland). Acetic acid was measured using an Acetic Acid Assay Kit (ACS Manual Format) from Megazyme (Bray, Ireland). Glycerol was measured using a Glycerol Assay Kit from Sigma-Aldrich (St. Louis, MO, USA). Ethanol was measured using an Ethanol Assay Kit from Sigma-Aldrich (St. Louis, MO, USA). The total protein content was measured using the Bradford Reagent assay protocol supplied by Sigma-Aldrich (St. Louis, MO, USA).

### 2.4. Amino Acid Analysis

All of the reagents, standards and solvents were purchased from Sigma-Aldrich (St. Louis, MO, USA) and used without further purification. The water was obtained from a Milli-Q filtration system (Millipore Filter Cor., Bedford, MA, USA) and degassed before use. The amino acid analyses were performed with a 1260 infinity Agilent HPLC (Agilent, Palo Alto, CA, USA) equipped with a 1260 DAD and FLD detector. Both the online derivatisation and instrumental method used were based on the Agilent 5991–5571 application note, with some modifications. The modifications include the calibration range (0.1–50 mg/L) using Norvaline as the internal standard (20 mg/L), an LOQ of 0.05 mg/L, and an LOD of 0.015 mg/L.

### 2.5. Separation Testing: Pre-Culturing and Co-Culturing of the Microorganisms in the Multi-Membrane System

Three different co-cultures were tested using the multi-membrane system, namely yeast–yeast, yeast–microalgae, and yeast–bacteria: *Lachancea thermotolerans*, *Chlorella sorokiniana* and *Lactiplantibacillus plantarum* were paired with *Saccharomyces cerevisiae*. *Chlorella sorokiniana* and *Saccharomyces cerevisiae* were isolated from winery wastewater [16], and *Lachancea thermotolerans* (IWBT B038) and *Lactiplantibacillus plantarum* (IWBT-Y1240) were obtained from the culture collection of the South African Grape and Wine research Institute (SAGWRI) at the University of Stellenbosch.

Precultures of each species were grown as monocultures in their respective media (Table 2), in 50 mL volumes, until the cultures reached the mid-log phase. The cell concentrations were established using optical density measurements prior to their inoculation.

The precultures were inoculated at an $OD_{600}$ and $OD_{750}$ of 0.1 for the yeast and microalgae, respectively, into 1.8 L of autoclaved media, and then these vessels were placed within the autoclaved multi-membrane system. Each combination of microbial species was performed in triplicate.

**Table 2.** Preculture and co-culture media for the multi-membrane system experiments.

| Species | Preculture Media | Optical Density Measurement | Co-Culture Media (with *S. Cerevisiae*) |
|---|---|---|---|
| *Saccharomyces cerevisiae* | Yeast extract-peptone-dextrose (YPD) medium | $OD_{600}$ | N/A |
| *Chlorella sorokiniana* | Tris-acetate phosphate (TAP) medium (Gorman and Levine 1965) containing $1\times$ Hom's vitamins (personal communication, Erik F. Y. Hom 2015) | $OD_{750}$ | Tris-acetate phosphate (TAP) medium (Gorman and Levine 1965) containing $1\times$ Hom's vitamins (personal communication, Erik F. Y. Hom 2015) with added 2% glucose |
| *Lactiplantibacillus plantarum* | De Man, Rogosa and Sharpe (MRS) agar | $OD_{600}$ | Yeast extract-peptone-dextrose (YPD) medium |
| *Lachancea thermotolerans* | Yeast extract-peptone-dextrose (YPD) medium | $OD_{600}$ | Yeast extract-peptone-dextrose (YPD) medium |

The growth of the microbial species was measured using hemocytometer cell counts every 12 h for 36 h for the yeast and microalgae co-culturing, and plate counts for the yeast and bacterial co-culturing. Samples were also plated out from the vessels onto WL, MRS and TAP media to ensure that no movement of species between the vessels or contamination had taken place. These plates were incubated at 25 °C for 3 days before counting.

*2.6. Yeast–Microalgae System with Indirect Contact*

The yeast–microalgae system was selected for further metabolic analysis to observe the production and utilization of various metabolites in an indirect contact experiment. The growth of both species was monitored using $OD_{600}$ and $OD_{750}$ for *S. cerevisiae* and *C. sorokiniana*, respectively. Samples were plated out every hour for the first 12 h, and every 6 h thereafter to check for any movement of partner species across the membrane.

The glucose concentrations were measured using the photometric determination of the glucose (mg/L) in the sample material based on the Enzytec™ (Daejeon, Korea) fluid glucose method, as performed on a Thermo Scientific Arena™ 20XT Analyzer. The glycerol concentrations were measured using the photometric determination of glucose (mg/L) in the sample material based on the Enzytec™ fluid glycerol method, as performed on a Thermo Scientific Arena™ (Monza, Italy) 20XT Analyzer.

**3. Results**

*3.1. Multi-Membrane System Design and Functioning*

A diagrammatical representation of the new system is shown in Figure 2. The liquid culture runs through the cylindrical membrane in the housing, and a stainless-steel needle valve on the other end creates increased pressure in the system. This pressure forces the liquid media through the 0.1 μm pores on the ceramic membrane, and cells are excluded from the media due to pore size. Culture media without cells runs directly into the other vessel. Cells excluded by the membrane are washed through by continuously pumped liquid media, and are returned to the initial vessel, thereby preventing the build-up of cells on the ceramic membrane. This process is mirrored in the other vessel.

This system may be run with various ceramic filter pore sizes, depending on the size of the species cultured, and the vessels may be any volume larger than 300 mL. Both vessels

are placed on magnetic stirrers, and are continuously mixed throughout the culturing. In order to prevent the overflow of the vessels, the removal tube in one of the vessels is placed at a specified height in the reactor so that the media will not be removed from the vessel after a certain volume. In case of the build-up of cells on the membranes, the system can be backflushed by inverting the direction of the flow for one hour every six to twelve hours. The vessels were equipped with media-out, media-in, headspace sharing and cell-free culture media inlets from the partner vessel ports. A sample port was added to allow for the sterile sampling of both vessels during the co-culturing.

When co-culturing yeast and microalgae or two different yeast species, two 0.1 µm membranes are used on either vessel (Figure 2). When co-culturing yeast and bacteria, a second membrane is added to the bacterial vessel so that the liquid culture can run through a 0.1 µm membrane and then a 0.08 µm membrane in order to exclude bacterial cells (Figure 3). The membrane sizes were chosen based on the size of the microorganism being excluded. The second membrane was added to improve the separation of the bacterial cells from the media, and to reduce the pressure within the system. This was achieved by first running the media through a larger filter and then the smaller 0.08 µm filter to reduce the pressure on the smaller filter.

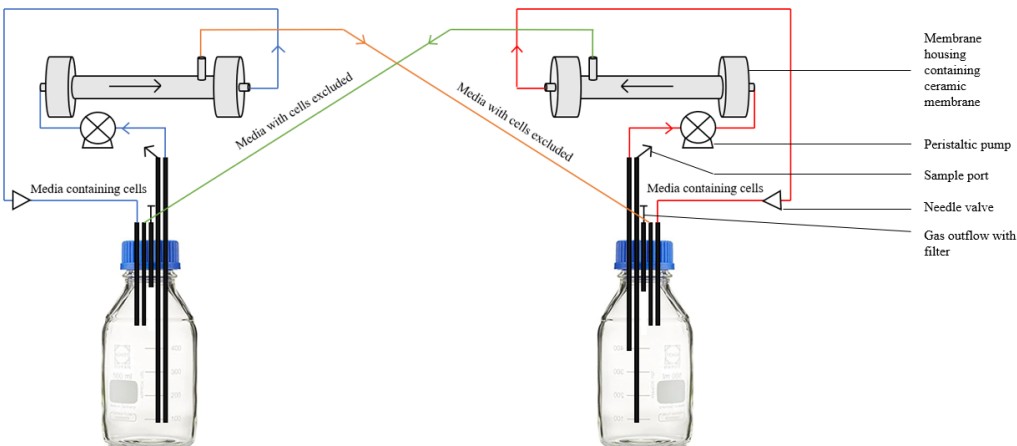

**Figure 2.** Schematic of the multi-membrane system setup for the yeast–yeast and yeast–microalgae studies using 0.1 µm membranes in both stainless steel housings.

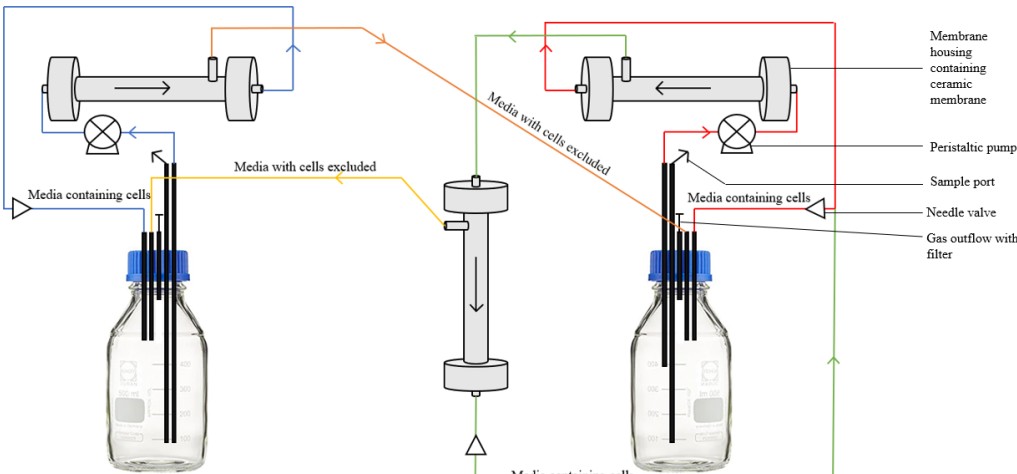

**Figure 3.** Schematic of the multi-membrane system setup for yeast–bacteria studies using a 0.1 µm membrane in one vessel and a 0.1 µm and a 0.08 µm membrane in the other.

### 3.2. Membrane Permeability Analysis of Various Metabolites

In order for effective co-culturing to take place, it must be ensured that a variety of metabolites can pass freely and quickly through the ceramic membranes while still preventing the transfer of whole cells between the vessels, without damaging the cells. This allows media free of cells to be passed between the vessels, and simulates an environment wherein the cells can exchange metabolic products without direct physical contact. In order to test this, one vessel was prepared to contain a variety of metabolites to simulate those present in media typical for the co-culturing of yeast, bacteria and microalgae, and produced by cell cultures such as sugars, proteins, amino acids, and alcohols, while the other vessel contained only distilled water. The system was set up to run over 8 h, and 10 mL volumes were sampled regularly at intervals of 10 min for the first 2 h, and every 30 min thereafter. These samples were stored at −20 °C until the metabolic assays could be completed.

These samples were tested using commercial enzyme assay kits to monitor the changes in the concentrations of their metabolites over time. The movement of the metabolites was analysed to ensure that the equal mixing of the vessels could take place during the co-culturing, and to determine the point of equalization in both vessels. The metabolites measured displayed a clear pattern of mixing between the two vessels, and equalization was shown to take place after 6 h of co-culturing, as seen in Figure 4A–E. The amino acids also showed equalization after 6 h of co-culturing at all of the concentrations seen in Figure 4F. It is important to note that no delay was present in the mixing of the larger protein molecules compared to the smaller molecules, such as the amino acids.

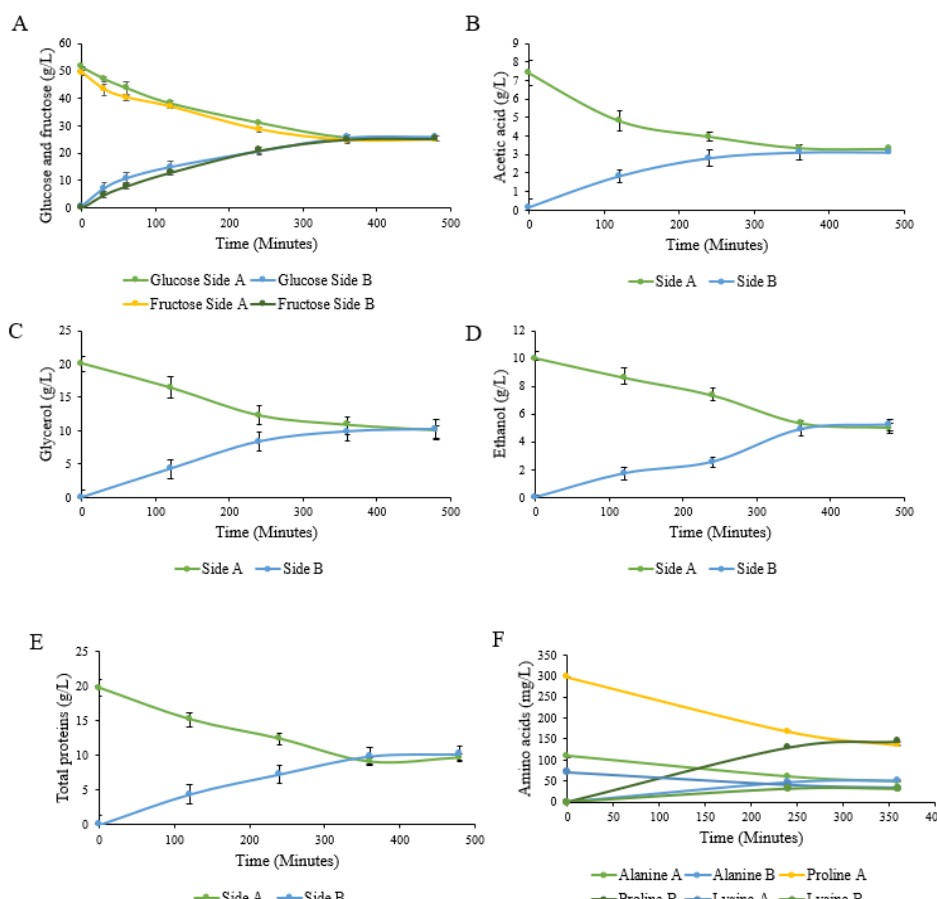

**Figure 4.** (**A**–**F**) Graph showing the change in concentration of (**A**) glucose and fructose, (**B**) acetic acid, (**C**) glycerol, (**D**) ethanol, (**E**) the total protein in g/L over time in the vessels A and B. (**F**) The change in concentration of the amino acids in g/L over time in vessels A and B. The data represent the mean ± standard deviation (*n* = 3).

### 3.3. Separation Testing of the Multi-Membrane System

The growth of microalgae, yeast and bacterial species was monitored using either hemocytometer counting techniques (yeast–microalgae system) or using plate counts (yeast–bacterial and yeast–yeast systems). The plates were incubated at 25 °C for 3 days before the counting took place. The plating out of the cultures also served as a contamination check for the system, and the culture samples were checked microscopically. Samples from the vessels were also plated out onto agar supporting the growth of the microorganisms in the partner vessel to ensure no mixing of cultures took place while the system was running. All of the species grown in indirect contact were able to proliferate after inoculation into separate vessels (Figure 5). The presence of the membrane and the pressure applied to the system did not prevent cell growth.

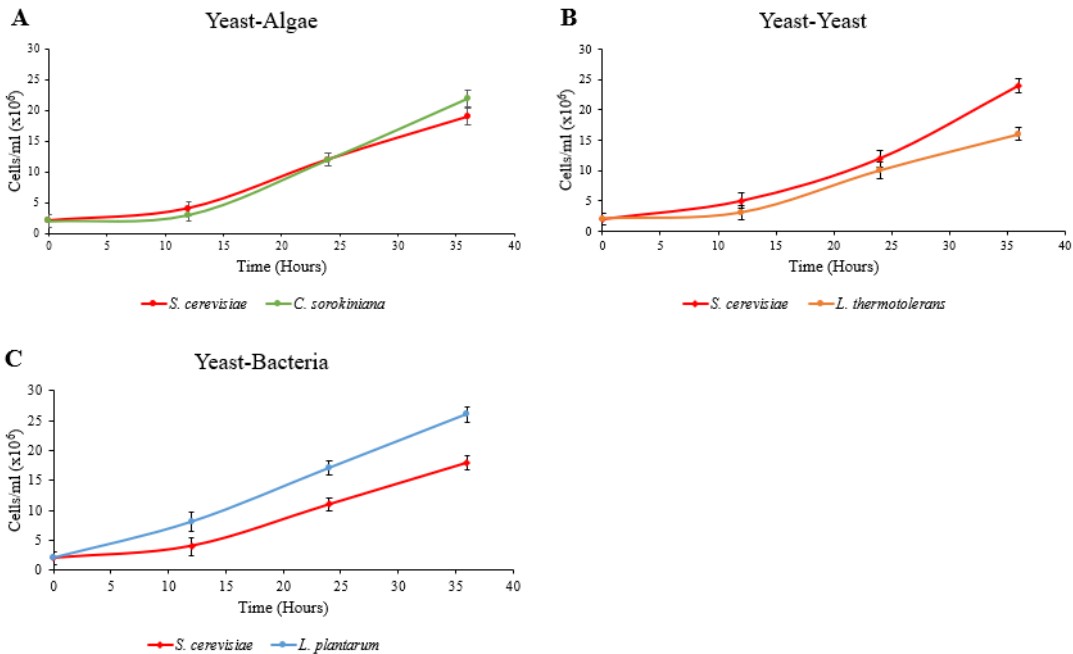

**Figure 5.** (**A–C**) Graph showing the growth of (**A**) *S. cerevisiae* and *C. sorokiniana*, (**B**) *S. cerevisiae* and *L. thermotolerans*, and (**C**) *S. cerevisiae* and *L. plantarum* in cells/mL over time in vessels A and B. The data represent the mean ± standard deviation (*n* = 3).

### 3.4. Yeast–Microalgae System with Indirect Contact between the Species

The yeast–microalgae indirect contact system underwent further characterisation to assess the efficient exchange of metabolites in co-culture conditions. Glucose, glycerol, ethanol, and acetic acid were monitored during the course of the co-culture. Samples were plated out throughout the co-culturing to ensure that no movement of species had occurred across the membrane (Figure 6).

*S. cerevisiae* and *C. sorokiniana* grew to an OD of between 2.7 and 3.5 over 36 h of co-culturing (Figure 7A). The growth rate of *C. sorokiniana* was consistent with that found in previous studies using this winery wastewater isolate. Glycerol (Figure 6B), produced by *S. cerevisiae*, increased over time, and the metabolite was efficiently exchanged between the two compartments, as the concentrations on both sides remained similar throughout the duration of the experiment, with only a small difference in the timing of the accumulation in line with the time required for the transfer from the producing compartment to the receiving compartment. Ethanol followed a similar pattern to glycerol, but a slightly larger difference in concentration, of 10–15%, between the two compartments was observed. This may be due to the continuous use of ethanol by *C. sorokiniana*, resulting in a continuous disequilibrium and a flow from the yeast to the microalgal bioreactor. This also highlights that no such system can ever be in a perfect balance, in particular when producer and user

organisms are co-cultured. Glucose was found at a concentration of 19 g/L in both vessels, and was utilized to a final concentration of approximately 15 g/L in both vessels after 36 h of co-culturing (Figure 7C,D). Glucose can be used a carbon source by both *S. cerevisiae* and *C. sorokiniana*, and the concentration remained consistent between the two vessels, implying that the sufficient mixing of the media was taking place.

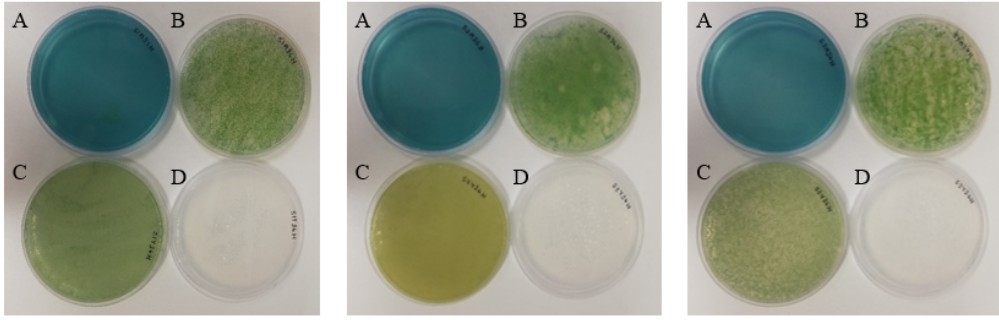

**Figure 6.** Images from the final 36-h sample point of the yeast–microalgae system, showing the plating out of undiluted samples onto WL agar (**A,C**) and TAP agar (**B,D**). The algae vessel samples were plated out onto plates A and B, and the yeast vessel samples were plated on plates B and C.

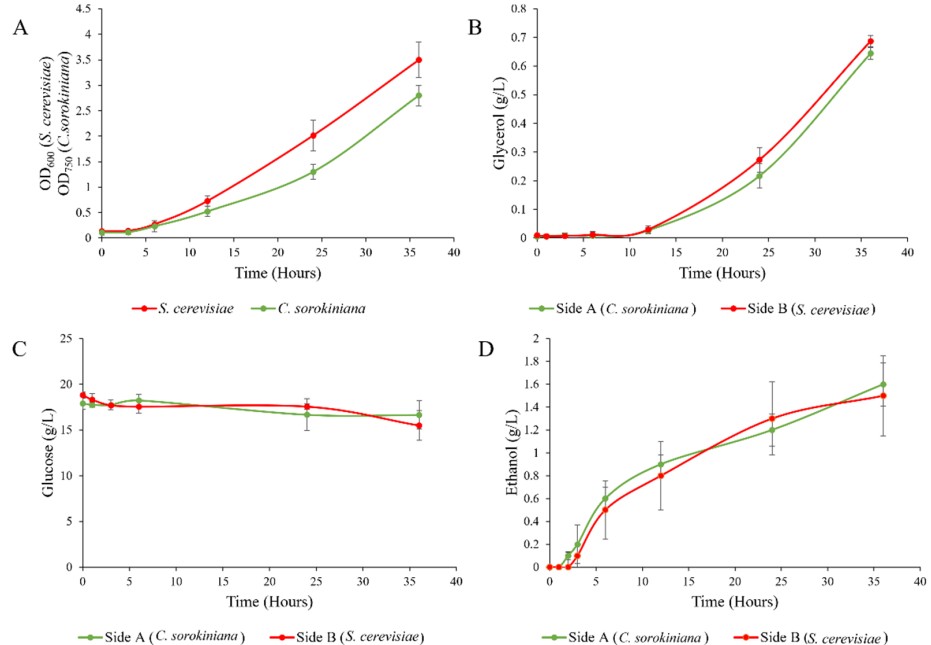

**Figure 7.** (**A–D**) Graph showing the $OD_{600}$ and $OD_{750}$ of *S. cerevisiae* and *C. sorokiniana* over 36 h of co-culturing (**A**). Graphs showing the concentrations in g/L of glycerol (**B**), glucose (**C**) and ethanol (**D**) in the 'Yeast' and 'Algae' vessels. The data represent the mean ± standard deviation (*n* = 3).

## 4. Discussion

The existing methods of studying interactions between co-existing species and the co-culturing of such species in physically separated compartments do not allow for a thorough investigation into how cell contact affects the nature of interaction. The ability to clarify the effects of metabolic contact in relation to cell contact between species will allow for the effective use of co-cultures and consortia in both industry and research. The system presented in this paper provides several improvements of existing technologies. The use of a cross-flow ceramic membrane system allows the flow-through of media and prevents the build-up of microbial cells on the membrane surface. A cross-flow membrane also exhibits limited shear on the cells, preventing damage to them. The multi-membrane

system utilizes the active flow of media from both vessels into the other, and complete mixing occurs after only 6 h of co-culture. Systems running at higher pressures may result in a quicker equalization of the media. The existing membrane co-culture technologies rely on either passive flow through a membrane or the alternation of the pumping of media through a membrane from either vessel, which result in much slower mixing times and inhomogeneous vessel compositions. This can create an uneven flow of metabolites throughout the co-culturing, and therefore fail to mimic the conditions in the natural environment. An added advantage of the current system is its extremely modular nature, which allows for the customization of the setup according to the experimental requirements; namely, the vessel size, flow and pressure, which can all be varied. This multi-membrane system also allows for larger liquid volumes to be sampled than the existing transwell systems for downstream analyses.

This study shows that this novel multi-membrane system efficiently and actively mixes media between two vessels using ceramic membrane filters, and that metabolic equalization occurs after only six hours using this experimental setup. The species are kept separate while sharing their produced metabolites and growth media, which is ideal for co-culture and consortia studies. Metabolites of various sizes and concentrations were tested, such as amino acids, sugars, and ethanol. The system could be run for a period of up to 36 h, and the pressure applied to force the media through the membrane did not prevent the proliferation of either species. A variety of species were used for the co-culturing, indicating the wide variety of applications this system could have. The system is inexpensive to construct, and is therefore accessible to most research applications. Most importantly, the novelty of the system lies in the active and continuous media-sharing between the two vessels that occurs during the co-culturing of the micro-organisms.

This study showed how these systems could be effectively be used in inter-species and inter-kingdom microbial studies. It should, however, be further investigated whether the application of pressure while driving the media through the membrane affects the growth of species within the multi-membrane system. The cell concentrations may be marginally affected by the system; however, the optical density reached by these species is in accordance with those seen in previous studies. It was noted that, upon the inspection of the undiluted plated-out samples from one of the indirect contact runs, two *S. cerevisiae* colonies were growing at the 12 h timepoint in the *C. sorokiniana* vessel. This meant that a few cells had likely crossed the membrane during the co-culturing; however, no more colonies were noted after 12 h, so it was assumed that *S. cerevisiae* had failed to proliferate in the alternate vessel. This did not occur every time the system was run, and no cross-over colonies were identified after 24 h in any multi-membrane system run. The pressure required of the system to push the media through the ceramic membrane is variable throughout the culturing, and studies with sugar concentrations higher than 40 g/L may require further optimization to prevent the leakage of the membrane.

Further research will focus on a variety of applications and alterations for investigation into the effects of microbial co-cultures and consortia. Investigations into the effects of co-culture at both the genotypic and phenotypic levels will assist in the further characterisation of the relationships between species in the natural environment Another advantage of this system is its scalability, as the individual bioreactors and the membrane surface size can easily be increased. The customizable element of the system also allows for changes to the membrane pore size for different species studies and the possible integration of a third vessel. Three-way consortia studies would require only the addition of another membrane, vessel, and valve. The system also holds potential for the treatment of liquids, with no required removal of species from the treated samples.

The system is currently patent pending; it holds great potential in the field of microbial co-cultures and consortia, and will assist in the further characterisation of a wide variety of microbial interactions for implementation in both research and industry.

## 5. Patents

A provisional patent, titled "A bioreactor for contactless co-culturing", has been filed in South Africa: Patent Application No.: 2021/06074.

**Author Contributions:** Conceptualization, F.F.B. and R.P.; methodology, R.P., D.R., F.F.B., J.R.O. and R.K.N.-B.; validation, J.R.O.; formal analysis, J.R.O. and R.K.N.-B.; investigation, J.R.O. and R.K.N.-B.; data curation, J.R.O.; writing—original draft preparation, J.R.O.; writing—review and editing, J.R.O., R.K.N.-B., D.R., F.F.B., R.P.; visualization, J.R.O.; supervision, F.F.B., R.K.N.-B., D.R. and R.P.; project administration, J.R.O., R.K.N.-B., D.R. and F.F.B.; funding acquisition, F.F.B. All authors have read and agreed to the published version of the manuscript.

**Funding:** This research was funded by the National Research Foundation (South Africa) through the South African Research Chair Initiative grant number (UID) 83471 to F.F.B., and the Royal Society (UK) through the Future Leaders: African Independent Researcher (FLAIR) Initiative to D.R.

**Institutional Review Board Statement:** Not applicable.

**Informed Consent Statement:** Not applicable.

**Data Availability Statement:** Data is available under the SUN Scholar data repository.

**Conflicts of Interest:** The authors declare no conflict of interest.

## Appendix A

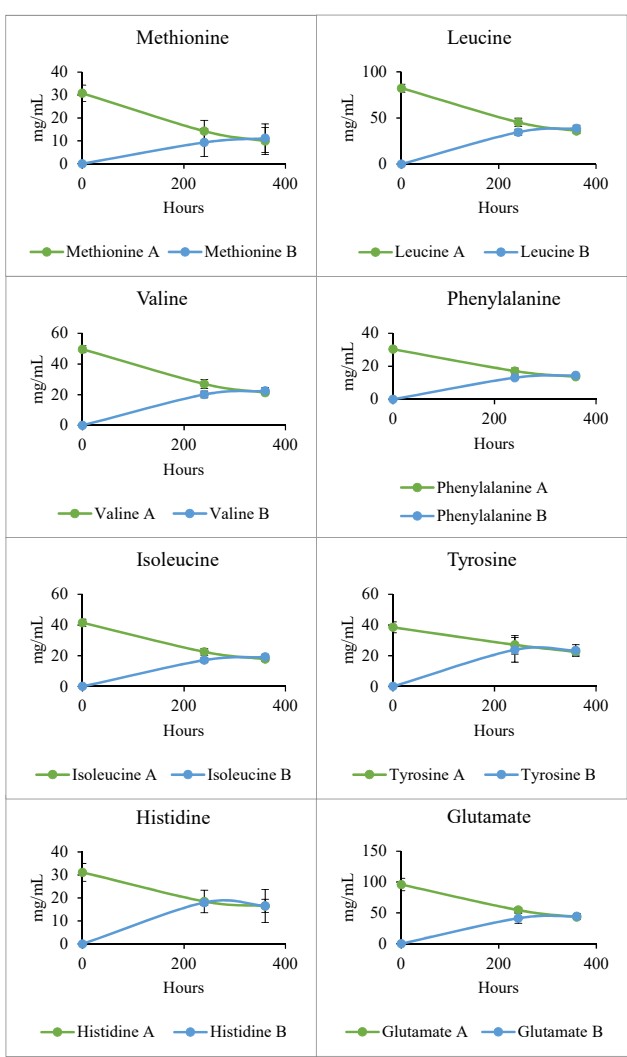

**Figure A1.** *Cont.*

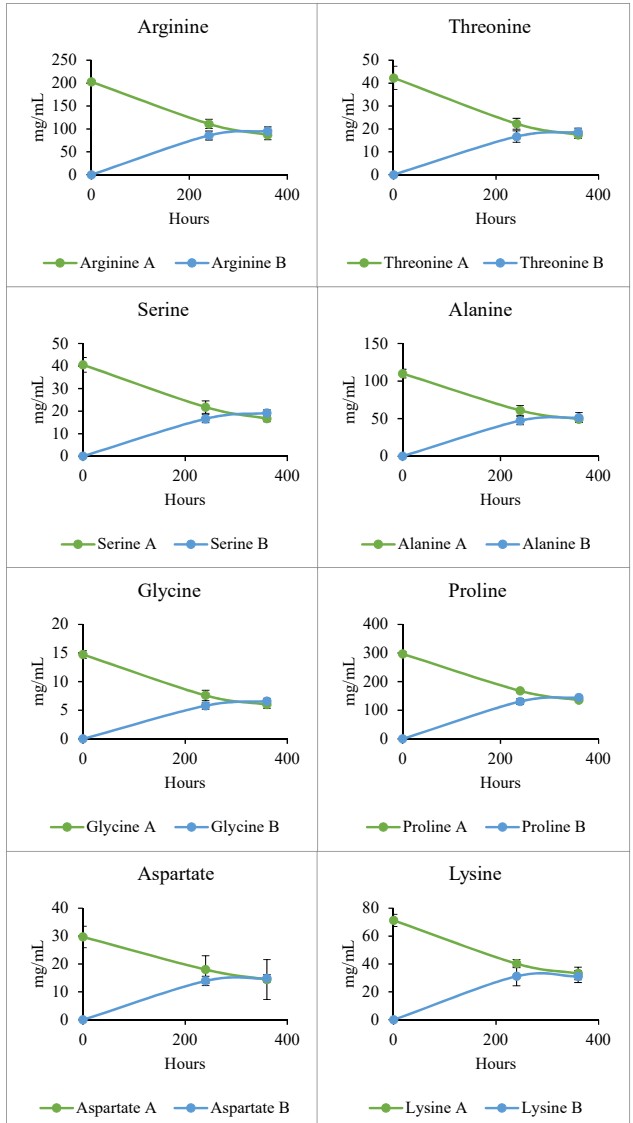

**Figure A1.** Graphs showing the change in concentration of amino acids in mg/mL over time in vessels A and B. The data represent the mean ± standard deviation (*n* = 3).

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
