# Peer review of "A Multi–Membrane System to Study the Effects of Physical and Metabolic Interactions in Microbial Co-Cultures and Consortia"

_fermentation, doi:10.3390/fermentation7040206_

Round 1

Reviewer 1 Report

The manuscript aimed to create and characterize a novel bioreactor system that uses ceramic membranes to create a multi-membrane compartmentalized system where two or more species can be co-cultured without mixing of species while ensuring efficient sharing of all media components. The authors concluded that this novel multi-membrane system efficiently and actively mixes media between two vessels using ceramic membrane filters
The authors presented in a clear and punctual way the data and the methodology used. But few minor corrections are necessary.
- Please use the updated bacteria nomenclature. Change Lactobacillus plantarum in Lactiplantibacillus plantarum
Figure 7. The microorganisms names are not in italics. Please correct the figure legend.
Together with reference 3 please cite also Perpetuini, G., Tittarelli, F., Suzzi, G., & Tofalo, R. (2019). Cell wall surface properties of Kluyveromyces marxianus strains from dairy-products. Frontiers in microbiology, 10, 79. 

Author Response

Thank you for the suggestions. The following changes were made:

              Updated the nomenclature to Lactiplantibacillus plantarum

              Changed microorganism names in figure 7 to italics

              Added reference Perpertuini et al., 2019.

              Fixed CO2 subindex

              Changed ml to mL

              References: added 2 updated references

Reviewer 2 Report

In a manuscript “A multi–membrane system to study the effects of physical and metabolic interactions in microbial co-cultures and consortia” authors describe device that allows for cultivating and investigating two microorganisms in a way, in which they are physically separated but share the cultivation media. I believe that this tool can be used to investigate the interactions between two organisms as a model for their interactions in nature.

The manuscript is well written. I only have several concerns that are listed below:

Line 93: parenthesis from the end of the sentence should likely correctly be placed after word ‘species’.

Line 125 Missing parenthesis after word China.

Fig. 1: I think that the drawing doe’s not provide the explanation of the device in a clear way. I believe that the drawing can be improved so that it is more clear what is what and where in A the membrane B really is.

Line 134: I don’t understand the last sentence in the Fig. 1 legend. It is likely an editing error.

Information on used strains of microorganisms should be provided (e.g. which strain of S. cerevisiae has been used).

Line 206: I believe there is mistake in the sentence: “Culture media containing cells media runs directly into the other vessel.“ I believe it should be “Culture media without cells runs directly into the other vessel.“

Line 234: In figure there is no vessel A and vessel B while these are mentioned in a legend.

Lene 251: “The metabolites measured display … “ should be “The metabolites measured displayed … “ as the results should be described in past tense.

Figure 7: (S.C.) and (C.S.) in axis description in panel A should be (S.c.) and (C.s.). If possible names of organisms in legends shoul be in italics.

Lines 351-353 Sentence “It must also be noted that during co-cultures 351 approximately 2 colonies from an undiluted 100 ul sample per run were noted to have 352 crossed the membrane after 12 hours, but none were noted thereafter ” is hard to understand I would suggest to reword the sentence.

Author Response

We made the changes as requested by the reviewers. Thank you for picking up those issues.

              Updated the nomenclature to Lactiplantibacillus plantarum

              Changed microorganism names in figure 7 to italics

              Added reference Perpertuini et al., 2019.

              Fixed CO2 subindex

              Changed ml to mL

              References: added 2 updated references

Reviewer 3 Report

I think that the masucript is interesting for scientist and researchers of the authors' field, and it is suitable for publication. There are just a couple of formal issues to correct: CO2 (2 should be as subindex; ml and mL should be written coherently, and numbers with decimals must be written with point instead of comma. On the hand, in relation to the content, in my opinion, some of the references used in the introduction and for the discussion, are somewhat old, and it would be recommendable to include some updated and more recent references to support both sections (introduction and discussion).

Author Response

Thank you for the repot and suggested changes, which were implemented as described below:

Updated the nomenclature to Lactiplantibacillus plantarum

              Changed microorganism names in figure 7 to italics

              Added reference Perpertuini et al., 2019.

              Fixed CO2 subindex

              Changed ml to mL

              References: added 3 updated references and re-numbered references in the text

              Changed figure 1 caption and image for clarity

              Added sourcing of microbial strains used

              Removed ‘vessel a’ and ‘vessel b’ from text

              Changed S.C and C.S to full species names in italics

              Rephrased lines 351 to 353

              Grammar changes: Added parenthesis (lines 93 and 125), rephrased (line 251)